# How long is a piece of loop?

Yoonjoo Choi[1], Sumeet Agarwal[2,3,4] and Charlotte M. Deane[5]

[1] Department of Computer Science, Dartmouth College, Hanover, NH, USA
[2] Department of Systems Biology Doctoral Training Centre, University of Oxford, Oxford, United Kingdom
[3] Department of Physics, University of Oxford, Oxford, United Kingdom
[4] Department of Electrical Engineering, Indian Institute of Technology Delhi, Delhi, India
[5] Department of Statistics, University of Oxford, United Kingdom

Corresponding author
Charlotte M. Deane,
deane@stats.ox.ac.uk

## ABSTRACT

Loops are irregular structures which connect two secondary structure elements in proteins. They often play important roles in function, including enzyme reactions and ligand binding. Despite their importance, their structure remains difficult to predict. Most protein loop structure prediction methods sample local loop segments and score them. In particular protein loop classifications and database search methods depend heavily on local properties of loops. Here we examine the distance between a loop's end points (span). We find that the distribution of loop span appears to be independent of the number of residues in the loop, in other words the separation between the anchors of a loop does not increase with an increase in the number of loop residues. Loop span is also unaffected by the secondary structures at the end points, unless the two anchors are part of an anti-parallel beta sheet. As loop span appears to be independent of global properties of the protein we suggest that its distribution can be described by a random fluctuation model based on the Maxwell–Boltzmann distribution. It is believed that the primary difficulty in protein loop structure prediction comes from the number of residues in the loop. Following the idea that loop span is an independent local property, we investigate its effect on protein loop structure prediction and show how normalised span (loop stretch) is related to the structural complexity of loops. Highly contracted loops are more difficult to predict than stretched loops.

## INTRODUCTION

Protein loops are patternless regions which connect two regular secondary structures. They are generally located on the protein's surface in solvent exposed areas and often play important roles, such as interacting with other biological objects.

Despite the lack of patterns, loops are not completely random structures. Early studies of short turns and hairpins showed that these peptide fragments could be clustered into structural classes (*Richardson, 1981*; *Sibanda & Thorton, 1985*). Such classifications have also been made across all loops (*Burke, Deane & Blundell, 2000*; *Chothia & Lesk, 1987*; *Donate et al., 1996*; *Espadaler et al., 2004*; *Oliva et al., 1997*; *Vanhee et al., 2011*)

or within specific protein families such as antibody complementarity determining regions (CDRs) (*Al-Lazikani, Lesk & Chothia, 1998*; *Chothia & Lesk, 1987*; *Chothia et al., 1989*). Loop classifications are generally based on local properties such as sequence, the secondary structures from which the loop starts and finishes (anchor region), the distance between the anchors, and the geometrical shape along the loop structure (*Kwasigroch, Chomilier & Mornon, 1996*; *Leszczynski & Rose, 1986*; *Ring et al., 1992*; *Wojcik, Mornon & Chomilier, 1999*).

Loops can also be classified in terms of function. There is some evidence that a loop can have local functionality. Experiments have been carried out which support the idea that swapping a local loop sequence for a different functional loop sequence enables the new function to be taken on (*Pardon et al., 1995*; *Toma et al., 1991*; *Wolfson et al., 1991*). One important example of functional loop exchange is in the development of humanised antibodies (*Queen et al., 1989*; *Riechmann et al., 1988*).

Accurate protein loop structure prediction remains an open question. Protein loop predictors have dealt with the problem as a case of local protein structure prediction. Protein structures are hypothesised to be in thermodynamic equilibrium with their environment (*Anfinsen, 1973*). Thus the primary determinant of a protein structure is considered to be its atomic interactions, i.e. its amino acid sequence. An analogous conjecture has arisen at the local scale where environment other than loop structure is fixed. Thus the modelling of protein loops is often considered a mini protein folding problem (*Fiser, Do & Sali, 2000*; *Nagi & Regan, 1997*). Although most loop structure prediction methods are based on this conjecture, loop sequence alone is not the complete determinant of the loop structure as even identical loop sequences can take multiple structural conformations depending on external environmental factors such as solvent and ligand binding (*Fernandez-Fuentes & Fiser, 2006*). Quintessential examples of such multiple loop structure conformations can be found in antibody CDR loops upon antigen binding (*Choi & Deane, 2011*).

Database search methods have been successful in the realm of loop structure prediction (*Verschueren et al., 2011*). They depend upon the assumption that similarity between local properties may suggest similar local structures. All database search methods work in an analogous fashion using either a complete set or a classified set of loops and selecting predictions using local features including sequence similarity and anchor geometry (*Choi & Deane, 2010*; *Fernandez-Fuentes, Oliva & Fiser, 2006*; *Hildebrand et al., 2009*; *Peng & Yang, 2007*; *Wojcik, Mornon & Chomilier, 1999*). Ab initio loop modelling methods aim to predict peptide fragments that do not exist in homology modelling templates without structure databases. Generally, ab initio methods generate large local structure conformation sets and select predictions (*de Bakker et al., 2003*; *Fiser, Do & Sali, 2000*; *Jacobson et al., 2004*; *Mandell, Coutsias & Kortemme, 2009*; *Soto et al., 2008*). The generated loop candidates are optimised against scoring functions. In all loop modelling procedures anchor regions are often problematic and the accuracy of loop modelling depends upon the distance between the anchors (*Xiang, 2006*).

Here, we focus on a specific local property of protein loop structure: the distance between the two terminal $C_\alpha$ atoms of the loop, which we refer to as its span. The nature of the span distribution is broadly similar across different protein classes or anchor types, except for loops linking anti-parallel strands (anti-parallel $\beta$ loops). In particular, the most highly frequent span appears to stay the same irrespective of the number of residues. This suggests that the span is distributed independently of other local properties and global structures. We demonstrate that the observed span distribution can largely be explained by a simple model of random fluctuations with a given length scale, based on the Maxwell–Boltzmann distribution.

It is widely believed that the accuracy of loop structure prediction depends on the number of residues, i.e. the larger the number of residues, the more difficult a loop is to predict (*Choi & Deane, 2010*; *Karen et al., 2007*). We introduce the normalised span which indicates how stretched a loop is (loop stretch $\lambda$). Fully stretched loops ($\lambda \simeq 1$) are almost always predicted accurately, whereas contracted loops ($\lambda \ll 1$) are harder to predict. In fact, shorter loops tend to be more stretched whereas longer loops are likely to be highly contracted. We suggest that loop stretch should be addressed in practical modelling situations and loop structure prediction should be concerned with predicting highly contracted loops.

## MATERIALS AND METHODS

### Loop definition

In each of the sets of protein structures loops, were identified using the following protocol. Secondary structures were annotated using JOY (*Mizuguchi et al., 1998*). A loop structure was defined as any region between two regular secondary structures that was at least three residues in length (*Donate et al., 1996*). Short (less than 4 residues in length) loops were discarded. Redundancy was removed using sequence identity. If a pair of loops shares over 40% sequence identity (*Fernandez-Fuentes & Fiser, 2006*), the loop which has a higher average B-factor was discarded.

### Membrane protein structures

Membrane proteins (3,789 chains) were extracted from PDBTM (*Tusnady, Dosztanyi & Simon, 2004*). The membrane layer was defined as being from $-20$ to $+20$ Å (*Scott et al., 2008*) from the centre of the protein and loops whose two end $C_\alpha$ atom coordinates were outside the layer were discarded. A total of 1,027 non-redundant membrane loops were defined.

### Soluble protein structures

All protein chains determined by X-ray crystallography which share less than 99% sequence identity ($<3.0$ Å resolution and $<0.3$ R-factor) were collected using PISCES (*Wang & Dunbrack Jr, 2005*) and all of our 3,789 membrane chains were removed. In order to get rid of any potential membrane chains in the list, PSI-BLAST (*Altschul et al., 1997*)

**Peer**J

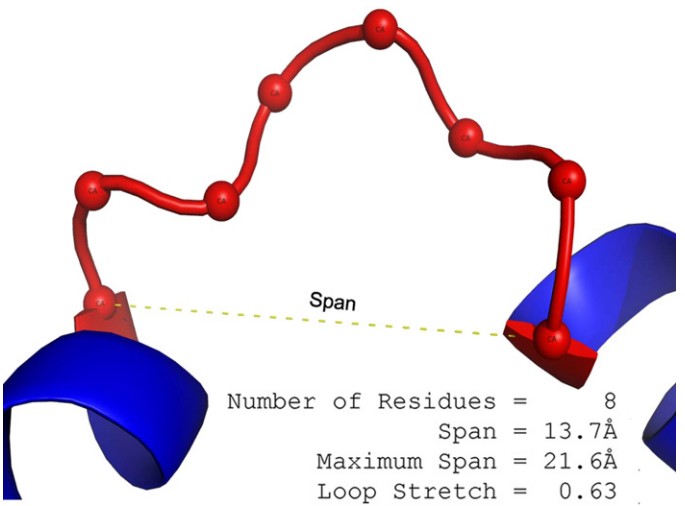

**Figure 1** **The definition of loop span and loop stretch.** Loop span is the separation of the two $C_\alpha$s at either end of the loop. In this example, 2J9O Chain A (198-205) has a span of 13.7 Å and contains 8 residues. Maximum span can be calculated from the number of residues in the loop to be 21.6 Å. Loop stretch is the normalised span (13.7/21.6 $\simeq$ 0.63).

was then used to compare the 3,789 membrane chains against the soluble set. Any chains found during 5 iterations with an E-value cut-off of 0.001 were removed from the list of soluble protein chains. A total of 25,191 non-redundant soluble loops were identified from 27,717 soluble protein chains.

## Loop span and loop stretch

The loop span ($l$) is the distance between the two terminal $C_\alpha$ atoms of a loop (Fig. 1).

The maximum span $l_{max}$ is a function of the number of residues $n$ and calculated as follows:

$$l_{max}(n) = \begin{cases} \gamma \cdot (n/2 - 1) + \delta & \text{if } n \text{ is even} \\ \gamma \cdot (n-1)/2 & \text{if } n \text{ is odd} \end{cases}$$

where $\gamma = 6.046$ Å and $\delta = 3.46$ Å (*Flory, 1998*; *Tastan, Klein-Seetharaman & Meirovitch, 2009*). If the distance between two terminal $C_\alpha$ atoms in the loop (i.e. the span) is $l$, the loop stretch ($\lambda$) of the loop is defined as a normalised span.

$$\lambda \equiv \frac{l}{l_{max}}. \tag{1}$$

Note that the values of $\gamma$ and $\delta$ are theoretical approximations so the $\lambda$ of some loops may occasionally be larger than 1. Similar notations are found in *Ring et al. (1992)*, *Tastan, Klein-Seetharaman & Meirovitch (2009)*.

# PROTEIN STRUCTURE PREDICTION AND LOOP STRETCH

## Loop modelling test sets

There are two modelling test sets. The first set includes loops of 8 residues. The loops were binned every 0.1 loop stretch. In each bin, 40 test loops were randomly selected. A total of 320 test loops from 0.2 to 1 in loop stretch were used (a full list is given in Table S1).

The second set consists of loops of between 6 and 10 residues in length. Two classes of loops were collected at each length: contracted loops ($\lambda < 0.4$) and stretched loops ($\lambda > 0.95$); an identical number of loops was kept in each of these classes at each length. A total of 346 test loops were identified (58, 72, 110, 58 and 48 loops respectively, See Tables S2 and S3). For example, there are 55 contracted test loops and 55 stretched loops for loops of 8 residues.

The measurement of accuracy is loop RMSD of all backbone atoms (N, $C_\alpha$, C and O) after superimposing anchor structures.

## MODELLER setting

The default loop refinement script was used. One hundred loop models were sampled under the molecular dynamics level of *slow*. The DOPE potential energy (*Shen & Sali, 2006*) was used for model quality assessment.

## FREAD setting

A database was constructed using the 27,717 soluble protein chains defined above. All the parameters were set as default (the environment substitution score cut-off value $\geq 25$). Any results from self-prediction were eliminated.

# RESULTS

## Nomenclature

In this paper, proteins are divided into two main classes: membrane and soluble proteins. Loops from membrane protein structures are called "membrane loops" and those from soluble protein structures are referred to as "soluble loops". Loops are also described by their secondary structure types: for example, loops connecting anti-parallel $\beta$ sheets are termed "anti-parallel $\beta$ loops". The physical spatial distance between the two end $C_\alpha$ atoms of a loop is referred to as "span" ($l$). Maximum loop span ($l_{max}$) is the furthest that a set of residues can spread. "Loop stretch" ($\lambda$) is the normalised loop span: the observed span between two $C_\alpha$ atoms at each end of a loop in a protein structure over the loops maximum span (Fig. 1).

## Loop span distribution

The number of residues in a loop is distributed in a similar fashion regardless of anchor types except for the loops linking anti-parallel $\beta$ sheets due to the constraint of hydrogen bonds between adjacent $\beta$ strands (Fig. 2A). Figure 2B displays how loop spans are

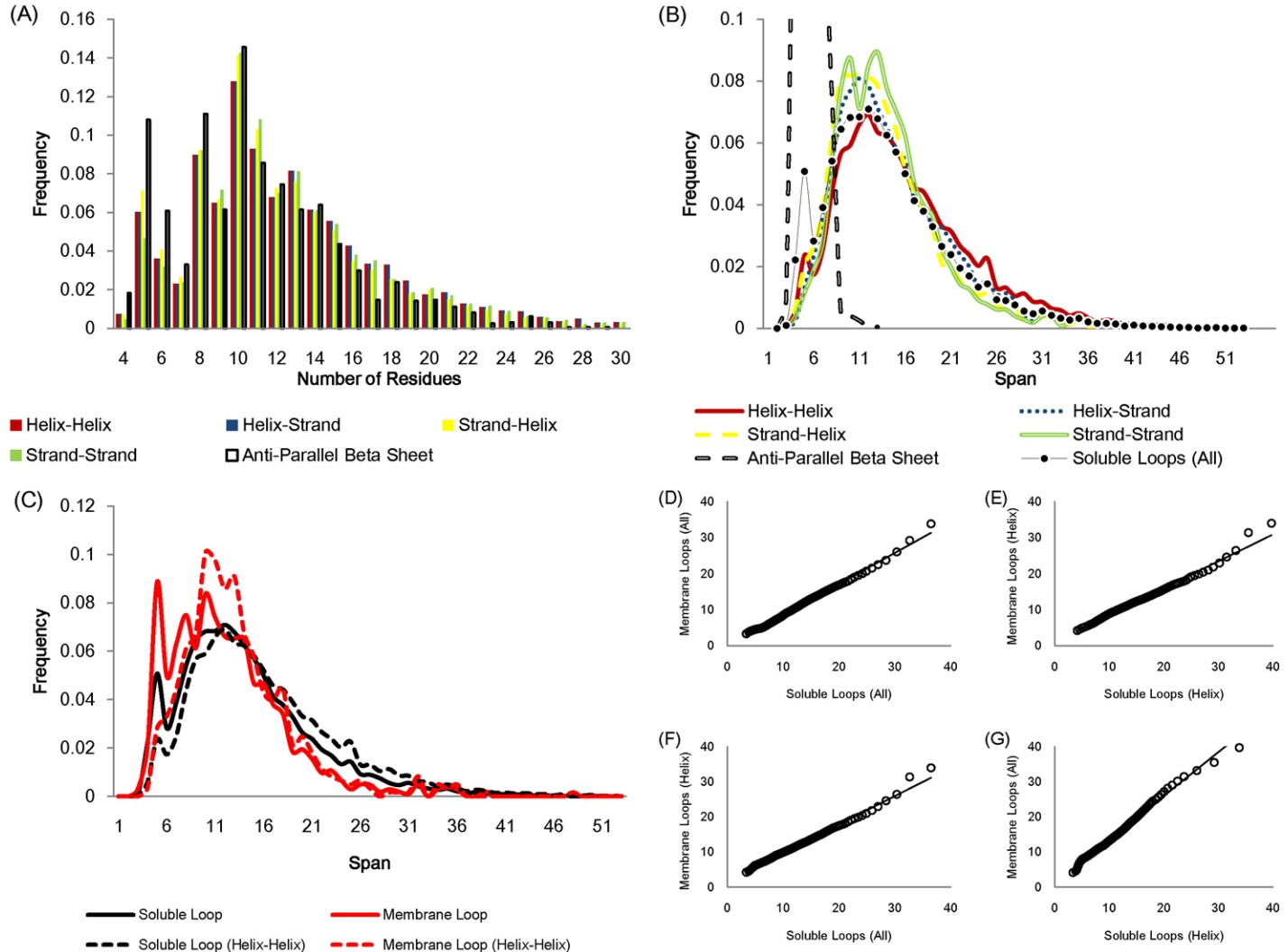

**Figure 2 Statistics of protein loops.** (A) The frequency distribution of loops containing different numbers of residues. Anti-parallel $\beta$ loops tend to have fewer residues. (B) The loop span distribution in terms of the anchor secondary structure do not show differences except for anti-parallel $\beta$ loops. The upper part of the anti-parallel $\beta$ loop span distribution is omitted in the figure. (C) The distributions of soluble loop span and membrane loop span appear to be similar. (D)–(G) Q–Q plots showing that the membrane and soluble loop span distributions are from the same probability distribution.

distributed for different anchor types. Again, apart from anti-parallel $\beta$ loops, the loop span distributions do not change with anchor structures.

The loop span distribution also does not alter when considering different protein classes. Figures 2C–2G show how the loop spans of membrane loops and soluble loops are distributed in a similar manner.

Essentially a loop span value reflects how distant the end tips of the two secondary structures that the loop connects are. These observations suggest that the loop span may be distributed independently of local anchor structures and protein types, i.e. anchor distances do not depend on local secondary structure elements or global protein structures.

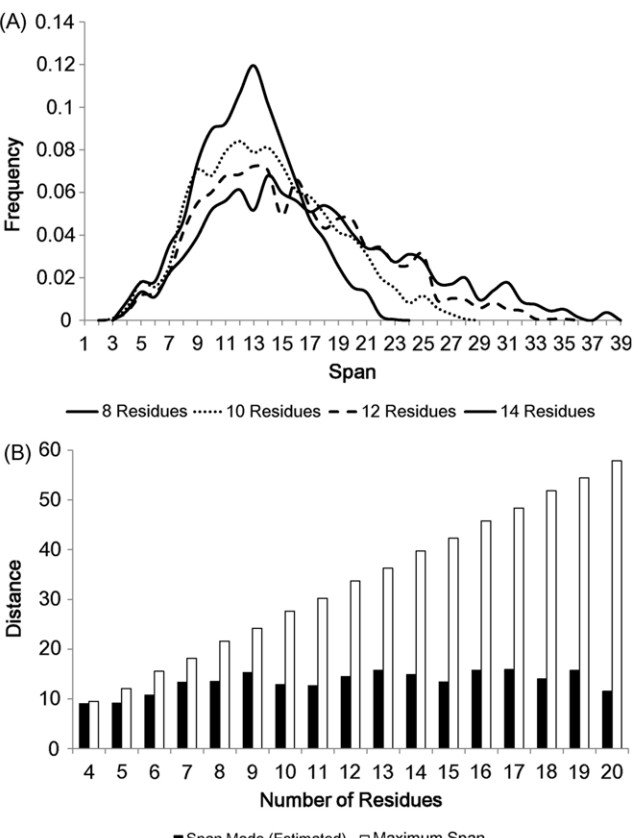

**Figure 3** **The span distributions for loops containing different numbers of residues.** (A) These appear to show a constant mode. Data here is soluble loops excluding anti-parallel beta loops. (B) The modes for the span distributions for loops containing different numbers of residues compared to the maximum span for that length. The span modes were estimated using the Gaussian kernel density estimation. Note that the estimated mode of loops of 4 residues is close to its maximum span.

The modes of loop span distributions are roughly constant (Fig. 2B), even if we split the loops in terms of the number of residues (Fig. 3A). We fit our data using the Gaussian kernel density estimation. The estimated distributions show a nearly constant mode ($\simeq 13$ Å on average, Fig. 3B). This constant span value may be due to protein packing. Folded proteins tend to be tightly packed and thus secondary structures are placed close to one another while avoiding side chain steric clashes. This packing effect may mean that the end points of two secondary structures (i.e. span) will lie within a constant span value regardless of the number of residues in a loop.

## Maxwell–Boltzmann distribution for loop span

From the above observations, it appears that loop span is distributed independently of local anchor structures or global protein classes. Here we assume that a protein loop is an independent unit of the protein structure and the span is determined regardless of any other effects including sequence or the rest of the structure.

Here a model for the loop span distribution is established under the hypothesis that the two end points of a loop fluctuate in three dimensional space, following the Maxwell–Boltzmann distribution. Two constraints are imposed in this model: the minimum span $l_{min}$ and the maximum span as a function of the number of residues $l_{max}(n)$. Within these constraints, the span oscillates according to a normal distribution $\mathcal{N}(\mu, \sigma^2)$ with a given length-scale $l_{mode}$ in three dimensional space.

The underlying assumptions are that the end points cannot approach each other too closely, and that there is a maximum span achievable for a loop with a given number of residues ($n$). Within these constraints, the span is allowed to fluctuate around the given length-scale $l_{mode}$ in three dimensional space. Thus, in this model, the loop span $l$ of $n$ residues is distributed as

$$l = \sqrt{l_x^2 + l_y^2 + l_z^2} \qquad l_x, l_y, l_z \sim \mathcal{N}\left(0, \frac{l_{mode}^2}{2}\right) \tag{2}$$

subject to the constraints that $l \geq l_{min}$ and $l \leq l_{max}(n)$, as stated above. The variance of $l_{mode}^2/2$ corresponds to a modal span of $l_{mode}$. Thus there are two parameters to be determined in our model: $l_{min}$ and $l_{mode}$. We set $l_{min}$ to 3.8 Å, which is the typical distance between two neighbouring $C_\alpha$ atoms in a protein chain. $l_{mode}$ is set to an estimate of the empirical mode using the Gaussian kernel density estimation (12.7 Å).

As there are not many longer loops in the data set, loops longer than 20 residues were discarded. In addition, all anti-parallel $\beta$ loops were eliminated due to their physical constraints. These eliminations left 21,597 soluble loops (The frequency distribution for each number of residues is in Fig. S2). Having set the two parameters $l_{min}$ and $l_{mode}$, loop spans were generated 10 times per model in accordance with the Maxwell–Boltzmann distribution, preserving the observed distribution of the number of residues (i.e. 10 simulated loop spans were generated for each real loop in the data set). The simulation outcome is depicted in Fig. 4A. The two distributions show the same shape and the quantile comparison in Fig. 4B indicates that they are statistically similar except for the tail region.

There are apparent anomalies between the simulated and real span distributions towards the extremes. The model seems to predict more short-span loops than observed. Our model imposes a sharp lower threshold at $l_{min} = 3.8$ Å, whereas in reality we expect a smoother transition. In other words, we expect our assumption of free fluctuation to break down when the span gets close to the lower bound and the physical constraints begin to become relevant. On the other side of the distribution, we see a substantially higher number of long-span loops ($>20$ Å) than predicted by the model. The mismatches in the long-span region tend to become more prominent as the number of residues is increased. When we examined which loops tend to have exceptionally long spans, we found that some of these "loops" are domain linkers between independent folding units and therefore likely to be under different constraints. Others appear to have been misclassified, as the loop definition used here is based only on the anchors containing at least three consecutive residues of secondary structures and the loop containing none. This allows segments such

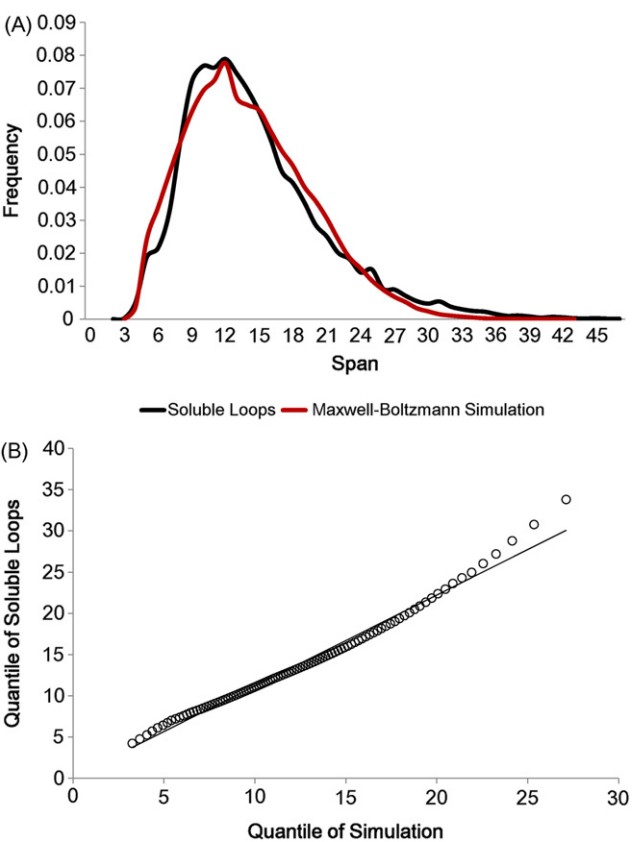

**Figure 4  Maxwell–Boltzmann distribution and loop span distribution.** (A) The loop span distribution (black) from soluble loops and that of the Maxwell–Boltzmann distribution (red). (B) The Q–Q plot suggesting that they follow the same distribution.

as termini structures to be included if there happen to be very short helical segments at a protein structure's terminus (Fig. S1).

## Protein structure prediction and loop stretch

The number of residues in loops is known to be related to the protein stability (*Nagi & Regan, 1997*) and the accuracy of most loop modelling techniques. Based on our observation that the loop span is independent of other properties, we examine its effects on protein loop structure prediction. Here we introduce loop stretch, the normalised loop span (Eq. (1)). Loop stretch values take on a range of 0 to 1, which indicates how stretched a loop is (1: fully stretched).

Figure 5 displays how loop stretch frequencies are distributed for different numbers of residues, demonstrating that the number of residues is negatively correlated with loop stretch, i.e. the longer a loop is, the more likely it is to be contracted. This may suggest that, instead of the standard belief that loop modelling performs worse as the number of residues in the loop increases, it may be that the real problem is better described by considering how stretched the loop to be predicted is. For example, if a loop contains many

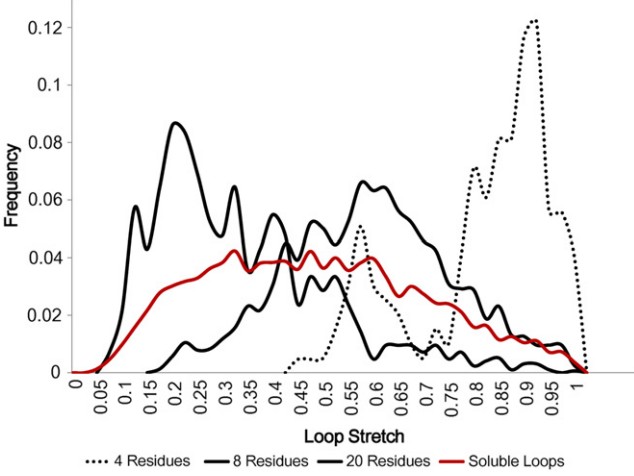

**Figure 5 Loop stretch of long and short loops.** Loop stretch distributions for loops containing different numbers of residues. Shorter loops tend to be more stretched whereas longer loops are likely to be more contracted. Only soluble loops excluding anti-parallel $\beta$ loops are plotted.

residues but is highly stretched, it will be predicted relatively accurately, as it can take on only a small number of different conformations.

In order to check the relationship between accuracy and loop stretch we used a test set containing only 8 residue loops with 40 non-redundant loops in every 0.1 loop stretch bin. Two loop modelling methods, which use two different sampling methods, were tested. MODELLER (*Fiser, Do & Sali, 2000*) is a popular protein structure prediction programme which has a built-in ab initio loop modelling module. FREAD (*Choi & Deane, 2010*) is a database search method which samples candidate loops depending on local properties and ranks predictions based on local loop sequence similarity and anchor geometry matches.

The average accuracy of MODELLER shows a negative linear correlation against loop stretch for the first test set (Fig. 6A). In the case of fully stretched loops ($\lambda > 0.95$), MODELLER can produce consistently accurate predictions, but its predictions worsen as the target loops are less stretched. FREAD produces more accurate predictions than MODELLER in general. However its predictions also begin to disperse as the loops become more contracted (Fig. 6B). FREAD generates candidate loops based on anchor matches and sequence similarity for a given loop target. This may imply that contracted loops tend to have multiple structural conformations or stringent sequence identity is required to predict such highly contracted loops. It should be noted that FREAD is not able to predict all the target loops due to the incompleteness of the structure database it uses (Fig. 6C).

In order to further assess the effect of loop stretch in loop structure prediction, MODELLER was re-examined on a second set. The second test set consists of loops from 6 to 10 residues in length. In this set, for each number of residues, the same numbers of loops (See Materials and Methods) were selected for both contracted ($\lambda < 0.4$) and fully stretched loops ($\lambda > 0.95$). MODELLER produces consistently accurate results for

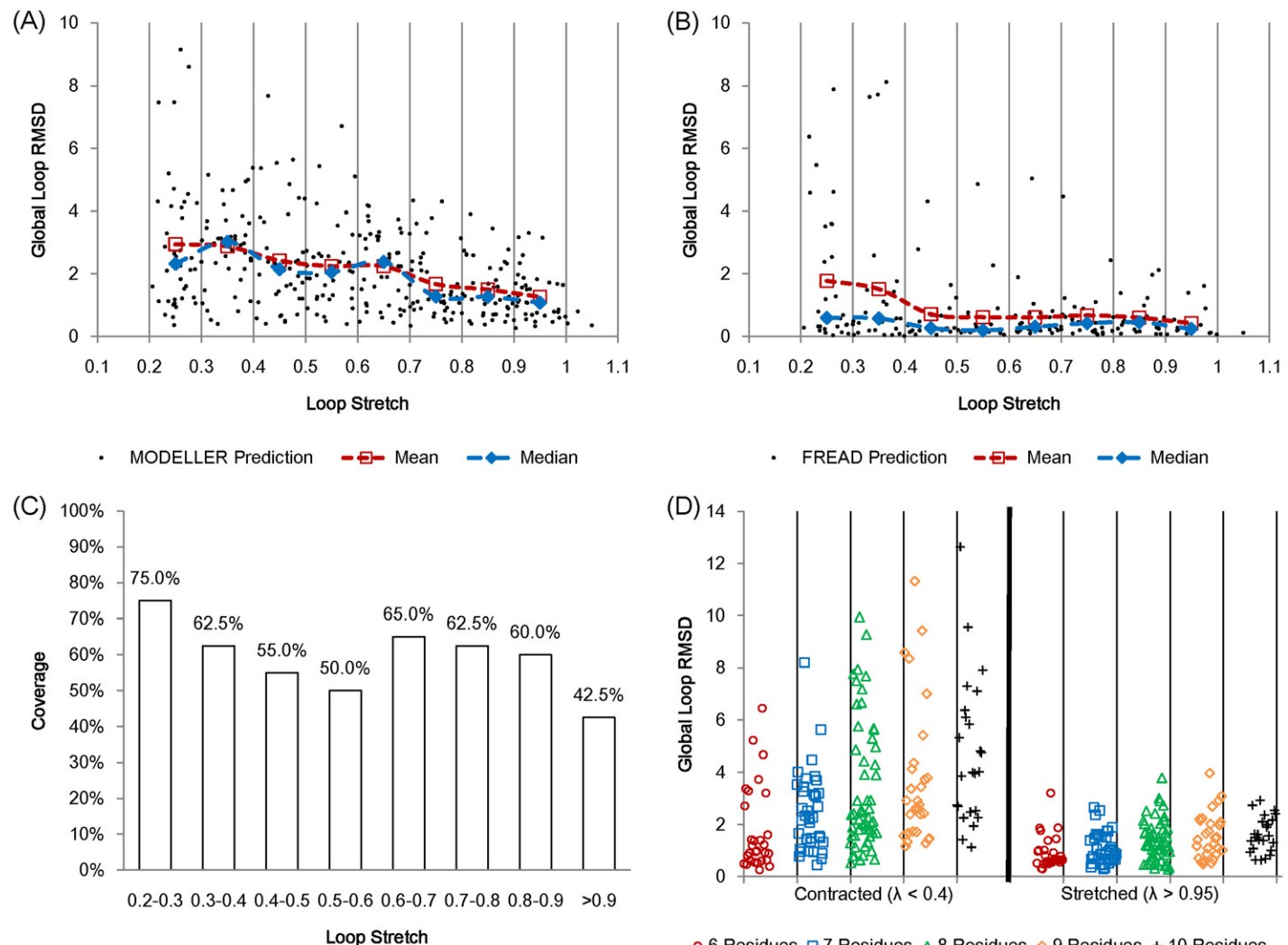

**Figure 6 Protein loop structure prediction and loop stretch.** Accuracy of protein loop structure prediction methods do not only depend on the number of residues, but also on loop stretch. MODELLER (A) and FREAD (B) both show accurate results when the target loop is stretched on the first set (including loops of 8 residues in length only). MODELLER shows worse prediction as loop stretch decreases whereas FREAD gives consistent accuracy on loop stretch. However both fail to predict very contracted loops ($\lambda < 0.4$). (C) The coverage of FREAD predictions in terms of loop stretch. (D) The second test set (contracted ($\lambda < 0.4$) and stretched ($\lambda > 0.95$) loops). The test loops are also split by the number of residues. For fully stretched loops ($\lambda > 0.95$), regardless of the number of residues, MODELLER predicts accurately.

fully stretched loops regardless of the number of residues, but fails to accurately predict contracted loops (Fig. 6D).

We calculated the partial correlations (Spearman's rank correlation) between accuracy, and the number of residues and loop stretch on the second test set so as to investigate what affects the prediction accuracy more (the number of residues or loop stretch). The partial correlation between loop stretch and RMSD is larger than that between the number of residues and RMSD ($-0.465$ and $0.367$ respectively). Loop stretch, just like the number of residues is something that can be calculated without knowledge of loop conformation and therefore can be used in the design of loop structure prediction software.

## DISCUSSION

In this paper, we focus on a specific local property (span) and demonstrate that the modes of loop span distribution appear to be independent of the number of residues. Loop span shows a distinct frequency distribution which does not depend on anchor types or protein classes. From these observations, we hypothesised that loop span is independent of the other effects and showed how the loop span distribution appears to correspond to a truncated Maxwell–Boltzmann distribution.

The reason behind the independence of loop span from the number of loop residues or secondary structure type is not known. The fact that the loop span distribution can be captured by a simple Maxwell–Boltzmann model allows one to speculate that protein loop structure prediction is indeed a local mini protein folding problem.

### Funding

Yoonjoo Choi was funded by the Department of Statistics, St. Cross College and the University of Oxford. Sumeet Agarwal was funded by the Clarendon Fund of the University of Oxford. The funders had no role in study design, data collection and analysis, decision to publish, or preparation of the manuscript.

### Grant Disclosures

The following grant information was disclosed by the authors:
Department of Statistics, St. Cross College and the University of Oxford.
Clarendon Fund of the University of Oxford.

### Competing Interests

Professor Charlotte M. Deane is an editor of PeerJ.

### Author Contributions

- Yoonjoo Choi conceived and designed the experiments, performed the experiments, analyzed the data, contributed reagents/materials/analysis tools, wrote the paper.
- Sumeet Agarwal performed the experiments, analyzed the data, contributed reagents/materials/analysis tools.
- Charlotte M. Deane conceived and designed the experiments, wrote the paper.

### Supplemental Information

Supplemental information for this article can be found online at http://dx.doi.org/10.7717/peerj.1.

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
