# Peer review of "How long is a piece of loop?"

_PeerJ, doi:10.7717/peerj.1_

## Round 0.1 · original submission · Minor Revisions

You manuscript needs minor revision.

·

Basic reporting

No comments

Experimental design

No comments

Validity of the findings

No comments

Additional comments

The manuscript of Choi et al. explores a specific property of protein loops: their span. It observes that loops with very different lengths (number of residues) can be fitted roughly to the same spans. They also describe a metric to asses the loop stretch, which in turn is informative to the quality of prediction: the more compact a loop, the more difficult to model it properly. As a consequence the authors argue that loop modeling methods should be tested and should be more focused on highly contracted loops, irrespectively of loop length, because these are much more challenging to model. Overall I found this work interesting and useful, the analysis is well done. I have a few minor comments:

(1) The Introduction describing the loop modeling approaches is slightly misleading. It is important to stress that loop conformations -once the modeling is treated as a mini protein problem - do not depend on the loop sequence only. As in case of full length proteins a given sequence folds in its native fold under physiological conditions. In case of full length proteins the physiological environment is the solvent (and temperature etc), while in case of loop it is the solvent AND the rest of the protein spanning it. As a consequence, as it was described by us (BMC Struct Biol. 2006 Jul 4;6:15.Saturating representation of loop conformational fragments in structure databanks. Fernandez-Fuentes N, Fiser A.) and others, loops with identical sequences can have different conformations depending on the protein they show up.

(2) In Fig 2C, I would guess that probably almost all membrane loops come from helix-helix connections, and this would require a comparison to helix-helix connected soluble loops and not to a mixed class, as it is now. In Fig2B helix-helix soluble loops spans are slightly shifted right for longer spans and on Fig2C membrane ones to the left, to shorter spans and therefore it is possible that once properly compared there will be a small but possibly significant difference.

(3) There might be some trivial explanation for the roughly constant span of loops: tightly packed proteins cannot afford large cavities and therefore the typical anchor distances of ends of secondary structures will be under strong thermodynamic selection. All loops , irrespectively of length will have to be accommodated within that given span. There are some references to it in the text, but I think it might be a driving effect.

(4) Fig. 6 illustrates one of the main points of the work that loop prediction accuracy is negatively correlated with loop span. While I do not doubt this effect, it seems that among compacted loops a few outliers contribute disproportionally to the worsening performance. I would suggest to use mean instead of average to account for the non-gaussian distribution more accurately.

·

Basic reporting

The paper meets all the criteria of PeerJ, it is sound, well written and it represents an important intelectual contribution to the field of loop structure prediction.

Experimental design

The authors do two things: they analyse some test sets of loops in terms of the stretch and span parameters and then evaluate loop prediction accuracy of the ab initio method Modeller to try to get an idea of difficulty of the structure prediction problem (judged by the accuracy of modeller's performance).

I like the concept a lot and it makes a lot of sense, complicated loop folds are essentially harder to predict - which still allows for longer loops to be harder than shorter ones, but length in not in itself the key determinant.

As an adept of database-based methods, I would have liked to see the same analysis using eg our freely available loopbrix method (http://brix.switchlab.org/), since this would tell if DB methods somehow have a different accuracy profile.

Validity of the findings

I think the study is well done and the results are clearly presented, the conclusions are supported by the data.

Additional comments

Cool paper.

---

## Round 0.2 · accepted · Accept

Paper accepted after modifications.